# Sequential Recommendation through Graph Neural Networks and Transformer Encoder with Degree Encoding

**Shuli Wang** [1] , **Xuewen Li** [2] , **Xiaomeng Kou** [2] , **Jin Zhang** [2,†] , **Shaojie Zheng** [2,†] and **Jinlong Wang** [2] and **Jibing Gong** [2,3,4,*]

1 School of Science, Yanshan University, Qinhuangdao 066004, China; wangshuli@ysu.edu.cn
2 School of Information Science and Engineering, Yanshan University, Qinhuangdao 066004, China; lixuewen@stumail.ysu.edu.cn (X.L.); kouxiaomengde@163.com (X.K.); nijzhang@163.com (J.Z.); shaojiezheng08@163.com (S.Z.); wangjinlong@stumail.ysu.edu.cn (J.W.)
3 The Key Lab for Computer Virtual Technology and System Integration of Hebei Province, Yanshan University, Qinhuangdao 066004, China
4 Key Laboratory for Software Engineering of Hebei Province, Yanshan University, Qinhuangdao 066004, China
* Correspondence: gongjibing@ysu.edu.cn
† These authors contributed equally to this work.

**Abstract:** Predicting users' next behavior through learning users' preferences according to the users' historical behaviors is known as sequential recommendation. In this task, learning sequence representation by modeling the pairwise relationship between items in the sequence to capture their long-range dependencies is crucial. In this paper, we propose a novel deep neural network named graph convolutional network transformer recommender (GCNTRec). GCNTRec is capable of learning effective item representation in a user's historical behaviors sequence, which involves extracting the correlation between the target node and multi-layer neighbor nodes on the graphs constructed under the heterogeneous information networks in an end-to-end fashion through a graph convolutional network (GCN) with degree encoding, while the capturing long-range dependencies of items in a sequence through the transformer encoder model. Using this multi-dimensional vector representation, items related to the a user historical behavior sequence can be easily predicted. We empirically evaluated GCNTRec on multiple public datasets. The experimental results show that our approach can effectively predict subsequent relevant items and outperforms previous techniques.

**Keywords:** sequential recommendation; graph neural networks; transformer encoder; degree encoding

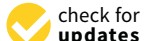



## 1. Introduction

Traditional recommendation algorithms either directly calculate the similarity between items in the data or calculate the similarity between users through the item, through techniques such as classic collaborative filtering [1] and matrix decomposition [2] The sequential recommendation can be viewed as learning a model in the time dimension to predict the items that users may interact with in the future through the items in the behavior sequence. Through the sequential recommendation model, the interest preferences of users can be more effectively captured according to the sequence of users' historical behavior, so that products that are more suitable for users can be recommended for users in a large number of goods and services. In the face of increasing data content, the data features become more and more sparse, and people's personalized demand for the platform is increasing. Traditional recommendation algorithms such as machine learning, collaborative filtering and sequential recommendation algorithms based on the Markov process and cyclic neural network have been unable to fulfill these needs.

Many existing efforts have been made towards user behavior understanding and concept extraction, such as course concept extraction [3], click-through rate prediction [4], and modeling user behaviors [5].

However, these approaches still suffer from two major limitations: (a) They ignore rich heterogeneous information in data. These approaches fully consider the semantic information of target items and leverage historical behaviors to make target item recommendation. Nevertheless, due to the few characteristics of the target item itself in some special data, potential semantic information cannot be mined. Therefore, it is critical to exploit the information hidden in the relationship between different entities in datasets; (b) They cannot consider multi-dimensional information simultaneously, such as considering the short-term and long-term preferences of users simultaneously, or capturing information from graphs and sequences structure at the same time. For example, although Yu Zhu et al. used time-LSTM to extract the features of the sequence while taking into account the information of the time interval between items in the behavior sequence, LSTM is not suitable for dealing with the problem of long sequences, so it ignores the user's long-term interest preferences. Therefore, as the sequence length grows, the impact of the previous items on the current item will become smaller or even disappear. Furthermore, it only considers the information contained in the sequence.

Here, we develop a novel framework GCNTRec to learn effective item representation in a user's historical behaviors sequence.

In our work, we construct graph structure data under different meta-paths between different entities, and discover richer node-related feature representations from the perspective of heterogeneous information networks. A heterogeneous information network is a special kind of information network containing multiple types of objects or multiple types of links. A heterogeneous information network can also be called a heterogeneous graph. The specific explanation of the heterogeneous graph is at the beginning of Section 3. It alleviates issues of data sparsity.

In order to make full use of the graph data information under the heterogeneous information network, GCN [6] is used to propagate the characteristic information of the neighboring nodes of the target node to obtain the final target node's vector representation, and combine with a user's historical behavior sequence to generate the user's vector representation.

On the basis of using graph neural networks to capture the user's short-term behavior preferences, the transformer encoder structure with a degree encoder is utilized for further learning the long-term interest preference contained in users' long behavior sequences. The transformer [7] encoder with the degree encoder not only combines the location information of the object nodes in the user historical behavior sequence, but also the unique graph data information to integrate the degree information of the item nodes in the graph.

We summarize our main contributions as follows:

(1) We propose a novel framework GCNTRec to learn effective item representation in a user historical behavior sequence. Specifically:

- Using GCN to extract features from graphs constructed under heterogeneous information networks can mine richer adjacency relationships between nodes.
- The transformer encoder with degree encoding intends to learn an object-rich sequence feature representation, which involves capturing long-term preference information and improving the quality of locating the contextual information of a sequence.

(2) The experimental results show that GCNTRec outperforms previous techniques on five widely used datasets.

The rest of this paper is organized as follows. Section 2 describes the proposed traditional recommendation algorithms and sequential recommendation algorithms. Section 3 describes the detailed design of our approach. Section 4 presents the evaluation results. Section 5 discusses our work. We conclude the paper in Section 6.

## 2. Related Work

### 2.1. Traditional Recommendation Algorithm

The traditional recommendation algorithm mainly includes the collaborative filtering algorithm based on similarity and the recommendation algorithm based on the double-tower structure of deep learning. The collaborative filtering algorithm, based on the essence of similarity including user attributes such as similarity, recommends to target users other items that similar users are interested in whilst taking into account the co-occurrence of items in the different users' history behaviors—thus obtaining item similarity and ultimately determining which target users interested in similar items to recommend to.The deep learning method is based on the neural network to learn the feature representation vectors of the user and the item. After combining the representation of the two vectors, the probability value of interaction between the two can be obtained through nonlinear functions. Finally, the probability value score is calculated and recommended to the item that the target user may be interested in.

### 2.1.1. Recommendation Algorithm Based on Collaborative Filtering

Collaborative filtering series of recommendation algorithms start their recommendation with the similarity among users or items. This means that similar users may have the same interests, or users may like items similar to the items they have purchased. Series of collaborative filtering algorithms can be divided into several categories, namely collaborative filtering based on user similarity, collaborative filtering based on article similarity and collaborative filtering based on model. Among these, user-based collaborative filtering was first proposed and implemented in the application of mail filtering. However, with the continuous development of website content, collaborative filtering based on user similarity also shows its own shortcomings. It requires a large amount of computation for each user to calculate and determine their similar users. The main feature of the user side in each major platform is constantly changing, and the item data are relatively fixed. At present, collaborative filtering based on user similarity is mainly used in scenes with drastic changes on the item side, such as news and information recommendation scenarios. Due to the shortcomings of user similarity, collaborative filtering algorithms based on item similarity have developed slowly. In contrast to collaborative filtering algorithms based on the user, collaborative filtering algorithms based on the item mainly combine different users' interactive item lists. The interaction here can include a variety of behaviors, such as purchasing, browsing, purchasing and collecting, and the similarity between different items can be measured by calculating the frequency of items appearing in the same user interaction list. Finally, the user's interest in items that have not been purchased or interacted with is calculated according to the user's interaction history items. After ranking the interest value, the item with the highest score can be recommended to the user. In addition to the above two types of collaborative filtering algorithms based on similarity of users or items, researchers have further proposed a model-based collaborative filtering algorithm. The typical representation of this kind of algorithm is the factorization machine (FM) [8], and a series of similar algorithms were derived from it, such as field-aware factorization machines (FFMs) [9] and factorization-machine-based neural network (DeepFM) [10] combined with deep learning technology.

### 2.1.2. Recommendation Algorithm Based on Deep Learning

Deep learning technology has made great achievements in the fields of natural language processing and computer vision. Recommendation algorithm researchers began trying to use deep learning techniques to enhance the effect of the item recommendation [11]. In the process of the research evolution of recommendation algorithms based on deep learning, a series of classic recommendation-related algorithms have emerged. Their emergence not only greatly improves the application effect of recommendation algorithms in production scenarios, but is also the basis of today's more refined, personalized and deeper research on recommendation scenarios. In 2016, Google proposed the wide

deep [12] model. On the one hand, it extracted the features of input data through manual feature engineering, and at the same time, it used a deep neural network to extract complex high-order features and combined the two features for prediction. DeepFM combines factorization and deep learning techniques, using factorization to extract low-order crossover features of input data and deep neural networks to extract the high-order crossover features of input data. At the same time, it combines low-level and high-level features into the multilayer perceptron to predict the probability of the user interacting with the item. For graph data in non-Euclidean space, the ACK [13] algorithm based on attention mechanism, combined with the graph convolution neural network, is also applied to the recommendation algorithm of a large-scale open learning platform.

*2.2. Sequence Recommendation Algorithm*

Compared with traditional recommendation methods, sequence recommendation further combines the characteristics of user historical behavior sequence, aiming to extract user's interest preference information through user historical behavior sequence. A user historical behavior sequence contains the context relationship between items [4,14], which is unique personalized information generated based on user's personal interest preference. Therefore, sequence recommendation can make full use of user behavior sequence to better meet the personalized needs of target users, and recommend the corresponding items by learning the interest preference contained in their behavior sequence [15].

2.2.1. Sequence Recommendation Algorithm Based on Markov Chain

Observing the user's behavior data through the Markov model to construct a state transition matrix can predict the user's behavior at the next point in time based on the state transition matrix. Thus, user data can be used more effectively, which is the reflection of the short-term interest and preference information of user behavior [12]. As a stochastic process, the subsequent state of the Markov chain is only related to its current state and has nothing to do with its past state. By analyzing the target user's behavior sequence, we can determine the probability of the target user operating the target item in the next time step under a certain behavior, which constitutes the state transition matrix of a user's interest preferences. Based upon the state transition matrix, we can predict the probability value of the target user's next behavior as an item. In order to better represent the user's state transition, Rendle et al. [1] proposed the method of weighted fusion of multiple different behavior sequences to generate a recommendation list.

2.2.2. Sequence Recommendation Algorithm Based on Recurrent Neural Network

In contrast to the sequence recommendation method based on the Markov process, thanks to the great achievements of the recurrent neural network (RNN) in the field of natural language processing, the recurrent neural network and its variants are mainly the long short-term memory network (LSTM) and gated recurrent unit (GRU). Both of these have been used in the case of the sequence recommendation algorithm. The recurrent neural network can capture the sequence relationship between the words in the text sequence and extract the characteristic information of the word context to generate the corresponding word vector representation. Consistent with the processing of text information by the recurrent neural network, the sequence recommendation treats the item nodes as words, and the user's historical behavior sequence can be regarded as a text sequence and input into the recurrent neural network, whilst the vector representation of each item in the user's behavior sequence can be generated through network learning. Similarly to the traditional recommendation method based on deep learning, Gru4rec [12] can integrate the vector representation of item nodes generated by the recurrent neural network to obtain the vector representation of user latitude, which can be used as the interest preference characteristics of target users. At the same time, combined with the vector representation of target items and based upon the twin tower structure, the probability score of target

users and target items can be finally obtained, respectively, and the recommendation list of target users can be further generated according to the score.

## 3. Our Model

The goal of our framework, GCNTRec, is to learn item representations in users' historical behavior sequences. The input to our model was an undirected finite heterogeneous graph G=(V,E), where V is a set of nodes and E is a set of edges. The heterogeneous graph is also associated with an object type mapping function $\Phi : V \rightarrow T^v$ and a link type mapping function $\varphi : E \rightarrow T^e$. Each node $v_i \in V$ has one node type, i.e., $\Phi(v_i) \in T^v$. Similarly, for $e_i \in E, \varphi(e_i) \in T^e$. When $|T^e| = 1$ and $|T^v| = 1$, it becomes a standard graph. The heterogeneous graph can be represented by a set of adjacency matrices $\{A_k\}_{k=1}^K$ where K = $|T^e|$ and $A_k \in R^{N \times N}$ is an adjacency matrix where $A_k[i, j]$ is non-zero when there is a k-th type edge from j to i. In our model, GCN [6] is used to learn the correlation between the target node and multi-layer neighbor nodes with adjacency matrices as input. Then, we use the transformer encoder to encode the item to obtain the second feature representation with long-term features. After that, we use the position encoding method provided by the transformer to encode the degree of each item node in the sequence. We concatenate the three vectors to derive the final sequence feature representation vector. Our model is shown in Figure 1.

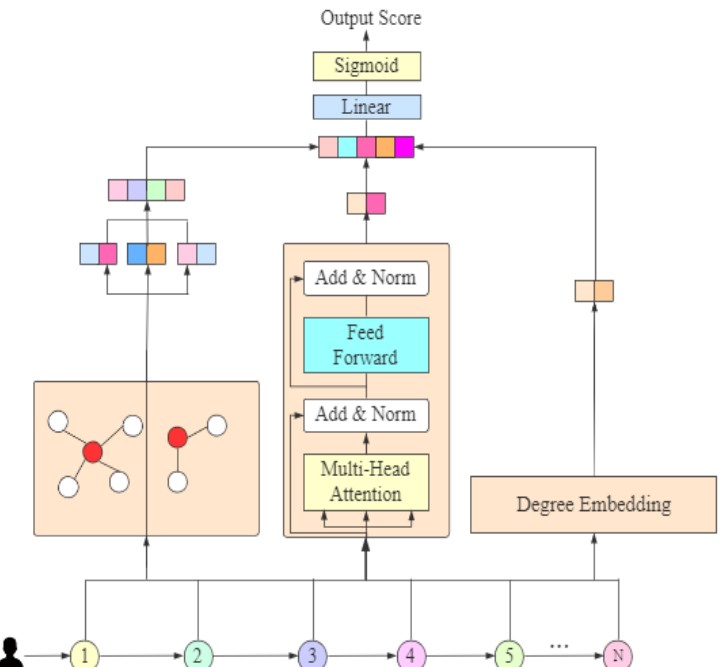

**Figure 1.** The structure of our model.

### 3.1. Item Embedding

We map each item in the dataset to a unique index. The index is embedded into a vector $x_e$ using Word Embedding:

$$x_e = \text{embedding (Size, dim)} \tag{1}$$

where $x_e \in R^{dim}$ is the embedding vector of a single item, Size is the total number of items in the dataset and dim is the dimension of the item vector. The feature vectors of each item are spliced together to form the final item embedding matrix X.

### 3.2. Graph Convolutional Network

Consider a meta-path set R = $\{r_1, r_2, \ldots, r_k\}$ comprising a sequence of k meta-paths, where k is the number of meta-paths in a dataset. The graph structure dataset correspond-

ing to the each meta-path is G = {$G_1$, $G_2$,..., $G_k$}. For one meta-path $r_k$, we use a two-layer GCN to embed $X \in R^{N \times dim}$ (Section 3.1) and graph data $G_k$ into a vector $e_k$ as follows:

$$H^0 = X \tag{2}$$

$$H_k^1 = \text{ReLU}\left(\tilde{D}^{-\frac{1}{2}}\tilde{A}\tilde{D}^{-\frac{1}{2}}H^0 W_k^1\right) \tag{3}$$

$$H_k^2 = \text{ReLU}\left(\tilde{D}^{-\frac{1}{2}}\tilde{A}\tilde{D}^{-\frac{1}{2}}H_k^1 W_k^2\right) \tag{4}$$

where $H_k^1$ and $H_k^2$, respectively, represent the first convolutional neural network layer output and the second convolutional neural network layer output, and $\tilde{A} = A + I$ where $A \in R^{N \times N}$ is the adjacency matrix of graph $G_k$, $I$ is the identity matrix, and $\tilde{D}$ represents the degree matrix of $A$. $X$ is thus transformed into a new feature matrix $e_k \in R^{N \times o}$ as follows:

$$e_k = H_k^2 \tag{5}$$

*3.3. Transformer*

3.3.1. Degree Encoding

The transformer incorporates nodes' location information to obtain the ability to capture sequence information between nodes. In our model, we used the positional encoding method in the transformer model to encode the degree of each node in the graph. The distribution of item degrees in the Book-Crossing dataset is shown in Figure 2.

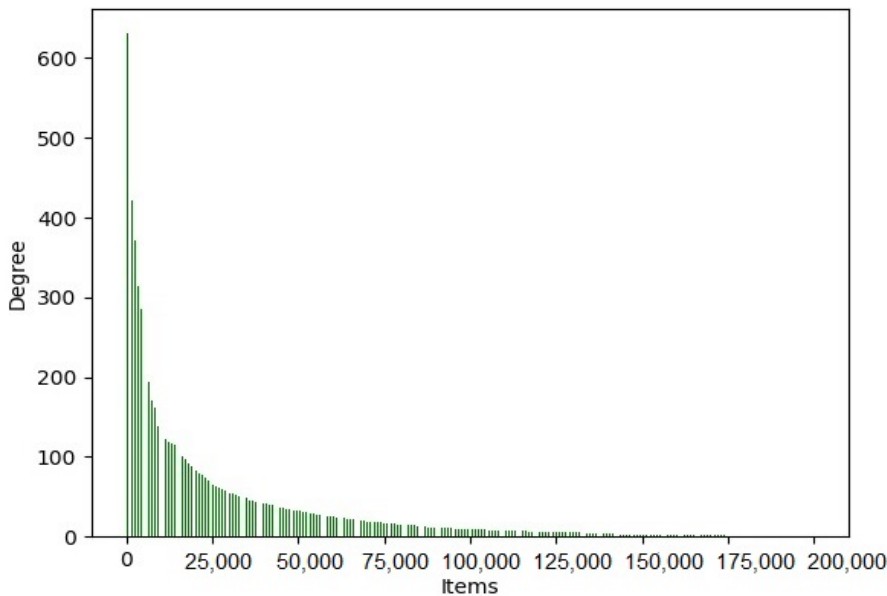

**Figure 2.** The distribution of item degrees in the Book-Crossing dataset.

Consider a user historical behavior sequence with $T$ items s = $(w_1,\ldots,w_T)$, and each item in sequence has its own degree value in the graph. The sequence of degrees, $d = (d_1,d_2,\ldots,d_t)$ where each value $d_i$ is embedded into a vector as follows:

$$x_{i+d_d} = \sin\left(\frac{d_i}{10,000^{\frac{2d_d}{D}}}\right) \tag{6}$$

$$x_{i+2d_d} = \cos\left(\frac{d_i}{10,000^{\frac{2d_d}{D}}}\right) \tag{7}$$

where $d$ represents the dimension of the embedded vector, $D$ represents the maximum degree value of all nodes in the graph. The degree embedding generated for all items in the sequence is represented by a matrix $X_n \in R^{T \times d}$.

### 3.3.2. Transformer Encoder

Given a user historical behavior sequence with $T$ items $s = (w_1, \ldots, w_T)$, we used the method mentioned in Section 3.1 to convert $s$ into an embedding matrix $X \in R^{T \times dim}$. In each attention head of the transformer encoder layer, $X$ first transformed into $Q$, $K$ and $V$, which represent three matrices as follows:

$$Q = XW^Q \tag{8}$$

$$K = XW^K \tag{9}$$

$$V = XW^V \tag{10}$$

where $W^Q$, $W^K$ and $W^V$ are parameter matrices with random initial values and $W^Q$, $W^K \in R^{dim \times d\_k}$, $W^V \in R^{dim \times d\_v}$. The self-attention matrix $Z$ is calculated by $Q$, $K$ and $V$ as follows:

$$Z = \text{Attention}\,(Q, K, V) = \text{softmax}\left(\frac{QK^T}{\sqrt{d_k}}\right)V \tag{11}$$

where $K^T$ is the transposed matrix of $K$. The transformer uses a multi-head attention mechanism. We obtain the multi-head attention matrix $Z_h ead$ as follows:

$$Z_{\text{head}} = \coprod_{h=1}^{\text{head}} \text{Attention}\,(Q_h, K_h, V_h) \tag{12}$$

At last, the multi-head attention matrix $Z_{head}$ is converted into representation vectors through the feedback layer in the transformer structure as follows:

$$e_T = \sigma(Z_{\text{head}}\,W_1 + b_1)W_2 + b_2 \tag{13}$$

where $e_T \in R_{T \times dim}$, $W_1$ and $W_2$ are the parameters' matricesand $b_1$ and $b_2$ are the bias. We concatenated the item feature matrix generated by the graph neural network $e$, the degree encoding matrix $X_n$ to $e_T$, to obtain the final item feature matrix $e_{all}$ as follows:

$$e_{all} = \coprod(e, e_T, X_n) \tag{14}$$

### 3.4. User Feature Representation

In the sequence recommendation task, the model predicts the probability score of the interaction between the target user and the target item by the nonlinear fusion of the two representation vectors. Given a user historical behavior sequence with $L$ items $s = (w_1, \ldots, w_L)$, the user's feature representation vector is calculated as follows:

$$u_i = \text{Maxpooling}\left(\coprod_{i=1}^{L} e_{all_i}\right) \tag{15}$$

where $L$ is the length of the sequence, and $e_{all_i} \in R^{o+dim+d}$ represents the i-th item's vector representation under our network mentioned in 2.3. $\coprod$ represents a concatenation operation. We integrate the vector representation of a series of users and a series of target items to generate a comprehensive feature representation vector as follows:

$$\text{emb}_{u_i e_j} = \coprod_{i=1}^{M} \coprod_{j=1}^{N} Linear\left(u_i e_{all_j}\right) \tag{16}$$

where $M$ is the number of user historical behavior sequences, $N$ is the number of items, $u_i \in R^{o+dim+d}$ represents the user's feature representation vector, $e_{all_j} \in R^{o+dim+d}$ represents the item's feature representation vector, and $\coprod$ represents a concatenation operation. In order to obtain the final probability score of the interaction between the target user and the target item, the feature vector needs to pass through a multi-layer perceptron structure to achieve nonlinear transformation and obtain the final target probability score through the sigmoid activation function.

$$score_{u_i e_j} = \text{sigmoid}\left(emb_{u_i e_j}\right) \tag{17}$$

where the score is the probability value of interaction between the target user vector and the target item vector.

The algorithm framework is shown in Algorithm 1.

---

**Algorithm 1** Sequential recommendation algorithm based on graph neural network and improved transformer.

---

**Input:** The set of meta paths $R$; The set of graph data based on meta-path $G$; Item feature vector $X$; The adjacency matrix corresponding to the graph data $A$; The degree matrix corresponding to the graph data $D$; Target user behavior sequence $L$; Number of training cycles *EPOCHS*.

**Output:** $W, b$ model related parameters.

1: **for** each *epoch* in *EPOCHS* **do**
2:     **for** *each $k$ in $R$* **do**
3:         $\tilde{A} \leftarrow A + I$
4:         $\tilde{D} \leftarrow \text{diag}(\tilde{A} \lessdot)$
5:         $H^0 \leftarrow X$
6:         $H_k^1 \leftarrow \text{Re}\,LU\left(\tilde{D}^{-\frac{1}{2}}\tilde{A}\tilde{D}^{-\frac{1}{2}} H^0 W_k^1\right)$
7:         $H_k^2 \leftarrow \text{ReL}\,U\left(\tilde{D}^{-\frac{1}{2}}\tilde{A}\tilde{D}^{-\frac{1}{2}} H_k^1 W_k^2\right)$
8:         $e_k \leftarrow H_k^2$
9:         Add $e_k$ into $E$
10:     **end for**
11:     **for** each $e_k$ in $E$ **do**
12:         $\alpha_k \leftarrow \frac{\exp(\sigma(e_k))}{\sum_{r \in R} \exp(\sigma(e_k e_r))}$
13:     **end for**
14:     $e \leftarrow \sum_{k=1}^R e_k \alpha_k$
15:     $D_{all} \leftarrow []$
16:     **for** each $D_i$ in $D$ **do**
17:         $D_{all} \leftarrow D_{\text{all}} + D_i$
18:     **end for**
19:     $X_n \leftarrow []$
20:     $X_{n_{i+d_d}} \leftarrow \sin\left(\frac{d_i}{10000^{\frac{2d_d}{D}}}\right)$
21:     $X_{n_{i+2d_d}} \leftarrow \cos\left(\frac{d_i}{10000^{\frac{2d_d}{D}}}\right)$
22:     $Q \leftarrow XW^Q$
23:     $K \leftarrow XW^K$
24:     $V \leftarrow XW^V$
25:     $Z_{\text{head}} \leftarrow \coprod_{h=1}^{\text{head}} \text{Attention}(Q_h, K_h, V_h)$
26:     $e_T \leftarrow \sigma(Z_{\text{head}} W_1 + b_1) W_2 + b_2$
27:     $e_{all} \leftarrow \coprod(e, e_T, X_n)$
28:     **for** each $l$ in $L$ **do**
29:         $u_i \leftarrow \text{Maxpooling}\left(\coprod_{i=1}^l e_{all_i}\right)$
30:         $emb_{u_i e_j} \leftarrow \coprod_{i=1}^M \coprod_{j=1}^N \text{Linear}\left(u_i e_{all_j}\right)$
31:         $score_{u_i e_j} \leftarrow \text{sigmoid}\left(emb_{u_i e_j}\right)$
32:     **end for**
33:     calculated loss $\leftarrow$ CrossEntropy $\left(score_{u_i e_j} - target_{u_i e_j}\right)$
34:     update parameters
35:     **if** *loss* not decline **then**
36:         break
37:     **end if**
38: **end for**
39: Return $W, b$
40: End

## 4. Experiments

### 4.1. Setup

#### 4.1.1. Datasets

We conducted our experiments on five public datasets, namely the MOOCs dataset, Book-Crossing dataset, Movielens-1m dataset, Steam-200k dataset and the Amazon Beauty dataset. The statistics of the four datasets are shown in Table 1.

The MOOCs dataset [16] is composed of the data of courses and student activities from the real environment of XuetangX. It contains the behavior sequence of users and their click-related knowledge points.

The Book-Crossing dataset [17] is composed of 278,858 user ratings in the Book-Crossing community. It contains 1,149,780 scores of approximately 271,379 books. The dataset contains three categories: users, books and book-rating. Books include the ISBN of the book and content-based information such as the author, publication year and publisher.

The MovieLens-1m dataset [18] contains 1 million ratings of 4000 movies by 6000 users. It is divided into three tables: rating, user information and movie information.

The Steam dataset [19] is a list of user behaviors with a large number of data on games played and purchased by players, as well as game titles, game prices and other information. The value indicates the degree to which the behavior was performed—in the case of 'purchase', the value is always 1; and in the case of 'play', the value represents the number of hours the user has played the game.

The Amazon Beauty dataset contains a large number of users' ratings of beauty products and timestamps with response scores.

**Table 1.** The statistics of the five datasets.

| Datasets | Users | Items | Records |
| --- | --- | --- | --- |
| MOOCS | 9986 | 1029 | 21,507 |
| Book-Crossing | 105,283 | 271,379 | 278,860 |
| Movielens-1m | 610 | 9742 | 108,375 |
| Steam | 12,393 | 5155 | 200,000 |
| Amazon Beauty | 1,210,271 | 1,249,274 | 2,023,070 |

#### 4.1.2. Metrics

We evaluated the sequential recommendation performance using five metrics, namely the AUC (area under the curve of ROC), accuracy, precision, recall and the F-Measure. These metrics were calculated through the confusion matrix. The confusion matrix is shown in Table 2.

**Table 2.** The confusion matrix.

| True | Predictions | |
| --- | --- | --- |
| - | Positive | Negative |
| Positive | True positive (TP) | False negative (FN) |
| Negative | False positive (FP) | True negative (TN) |

Accuracy refers to the ratio of correct prediction results to total samples:

$$\text{Accuracy} = \frac{TP + TN}{TP + FP + FN + TN} \tag{18}$$

Precision refers to the proportion of samples that are truly positive among all samples that are predicted to be positive:

$$\text{Precision} = \frac{TP}{TP + FP} \tag{19}$$

Recall refers to the proportion of samples that are actually positive that were predicted to be positive:

$$\text{Recall} = \frac{TP}{TP + FN} \tag{20}$$

The $F_1$-score is the harmonic mean of the precision and recall. Normally, precision and recall are contradictory to each other. Thus, in order to evaluate the quality of the model, $F_1$ is used as a comprehensive measure of precision and recall:

$$F_1 = \frac{2PR}{P + R} \tag{21}$$

where $P$ denotes precision and $R$ denotes recall.

AUC is defined as the area under the ROC curve (receiver operating characteristic curve). The abscissa of the ROC curve is the false positive rate and the ordinate is the true positive rate. AUC is used as a metric to visually compare which model has better generalization performance, and this metric will not be affected by the imbalance of the data category of the categorized data. The larger the value is, the better the generalization effect of the model is. The AUC calculation method is as follows:

$$AUC = \frac{\sum_{\text{ins}_i \in \text{ positive }} \text{rank}_{\text{ins}_i} - \frac{M \times (M+1)}{2}}{M \times N} \tag{22}$$

where $M$ represents the number of positive samples, $M$ represents the number of negative samples and $rank_{ins_i}$ is the ranking value of the predicted probability of the positive sample with the index of $ins_i$ in the predicted probability of all samples.

### 4.1.3. Baselines

We compared our sequential recommendation approach GCNTRec with three baseline methods.

PMF [20] is a traditional recommendation algorithm which starts with the correlation between users and items. It does not model user behavior sequence information.

GRU4Rec [21] is a sequential recommendation algorithm based on gate recurrent units. The use of gate recurrent units allows it to model user behavior sequences, and it combines multi-layer perceptron structure to learn the information contained in user behavior sequences.

Time-LSTM [5] is a sequential recommendation algorithm which considers time intervals based on long- and short-term memory. It can learn user behavior sequence information and consider the time interval information between items in the behavior sequence. This method further expands the relationship attributes between nodes in the sequence.

$GCNTRec_g$ is a variant of the GCNTRec which uses a graph structure to model data and uses a graph convolutional neural network to extract short-term behavioral sequence information between item nodes to generate the representation. Then, it generates the user's representation vector according to the user behavior sequence and finally generates user-to-items forecast results.

$GCNTRec_{gt}$ adds the encoder structure of the transformer on the basis of $GCNTRec_g$ to extract long-term behavior feature information in users' historical behavior sequence.

$GCNTRec_s$ is our full model that adds the encoder structure of the transformer on the basis of $GCNTRec_{gt}$ to extract long-term behavior feature information in users' historical behavior sequence.

### 4.1.4. Hyperparameters

We set the length of the user behavior sequence to 10. We train our models using the Adam optimizer with an initial learning rate of $10^{-4}$ and a weight decay of 0.0003. We set the mini-batch size and dropout rate to 512 and 0.1, respectively. We trained our models

for a maximum of 1000 epochs and performed an early stop if the validation performance did not improve for ten consecutive iterations.

For the transformer encoder, we set the number of layers and heads to 2 and 1, respectively. The input size and the hidden size were set to 128 and 256, respectively. In the degree encoder, we set the output dimension of the degree encoder to 32.

For graph neural networks, we used a two-layer GCN. We set the input dimension and output dimension of the first layer to 128 and 64, respectively, and the input dimension and output dimension of the second layer were set to 64 and 32, respectively.

### 4.2. Results and Analysis

4.2.1. Overall Results

The overall results of our proposed model and baselines are presented in Table 3. The results show that the base model outperforms most of the baselines, while the full model further improves the performance. Our full model $GCNTRec_s$ improves the performances by 0.094, 0.053, 0.038, 0.02, and 0.02 AUC points compared to the best results in the baselines for the MOOCs, Book-Crossing, Movielens-1m, Steam, and Amazon Beauty datasets, respectively. Furthermore, for the other metrics, our full model also surpasses previous best models by a significant margin. We used critical difference diagrams for all three metrics. The critical difference diagrams for AUC, precision and $F_1$ are shown in Figures 3–5. From these three critical difference diagrams, we can clearly see how on average our full model had the best algorithms over the five datasets

**Table 3.** Comparison of the results of each model.

|  | MOOCs | | | Book-Crossing | | | Movielens-1m | | |
|---|---|---|---|---|---|---|---|---|---|
|  | AUC | Precision | $F_1$ | AUC | Precision | $F_1$ | AUC | Precision | $F_1$ |
| PMF | 0.757 | 0.658 | 0.667 | 0.643 | 0.564 | 0.591 | 0.830 | 0.745 | 0.758 |
| GRU4Rec | 0.812 | 0.719 | 0.730 | 0.716 | 0.625 | 0.651 | 0.873 | 0.789 | 0.806 |
| Time-LSTM | 0.829 | 0.744 | 0.752 | 0.691 | 0.606 | 0.621 | 0.881 | 0.797 | 0.815 |
| $GCNTRec_g$ | 0.885 | 0.790 | 0.825 | 0.708 | 0.609 | 0.650 | 0.887 | 0.805 | 0.824 |
| $GCNTRec_{gt}$ | 0.904 | 0.801 | 0.841 | 0.725 | 0.634 | 0.668 | 0.896 | 0.813 | 0.836 |
| $GCNTRec_s$ | 0.923 | 0.828 | 0.858 | 0.769 | 0.685 | 0.712 | 0.919 | 0.835 | 0.859 |

|  | Steam | | | Amazon Beauty | | |
|---|---|---|---|---|---|---|
|  | AUC | Precision | $F_1$ | AUC | Precision | $F_1$ |
| PMF | 0.727 | 0.648 | 0.663 | 0.690 | 0.601 | 0.635 |
| GRU4Rec | 0.753 | 0.688 | 0.694 | 0.738 | 0.643 | 0.669 |
| Time-LSTM | 0.747 | 0.661 | 0.689 | 0.727 | 0.635 | 0.658 |
| $GCNTRec_g$ | 0.752 | 0.668 | 0.692 | 0.730 | 0.640 | 0.663 |
| $GCNTRec_{gt}$ | 0.766 | 0.684 | 0.708 | 0.741 | 0.652 | 0.676 |
| $GCNTRec_s$ | 0.773 | 0.692 | 0.717 | 0.758 | 0.674 | 0.698 |

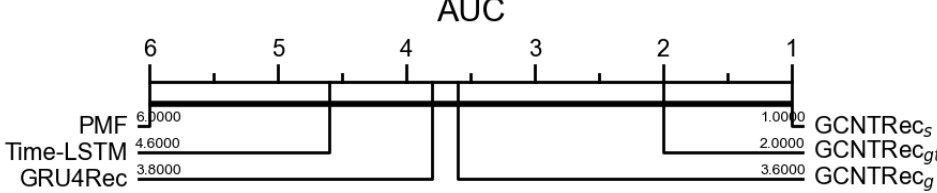

**Figure 3.** Critical difference diagrams for AUC.

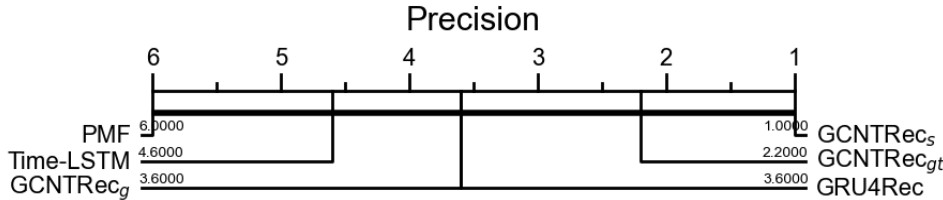

**Figure 4.** Critical difference diagrams for precision.

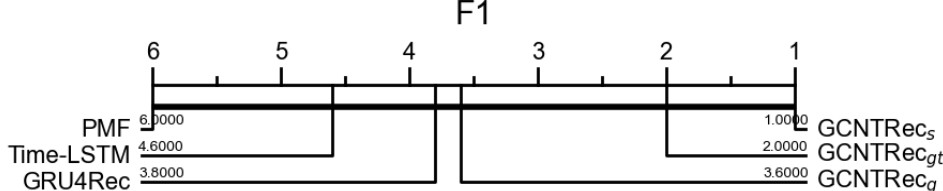

**Figure 5.** Critical difference diagrams for $F_1$.

GCNTRec and its variant methods based on the same dataset have different experimental results, and their results all roughly show the same trend, while $GCNTRec_g < GNTRec_{gt} < GNTRec_s$ and $GCNTRec_g$ has better results than other comparison algorithms in some datasets, reflecting that the graph convolutional neural network has a good extraction effect on the feature of the item. The result $GCNTRec_g < GCNTRec_{gt}$ shows that the model combined with the transformer encoder further extracts the implicit connections between items in the sequence. The $GCNTRec_{gt} < GNTRec_s$ shown in the experimental results further illustrates the improvement effect brought by adding the item degree encoding which makes the model have a stronger expressive ability to generate a more efficient node representation vector.

### 4.2.2. Effect of the Number of Transformer Encoder Layers

In this section, we explore the effect of the number of transformer encoder layers on the previously mentioned classification task metrics. We trained a number of GCNTRec models with a differing number of layers, while otherwise using the same hyperparameters and training procedure as previously described. Results in the Book-Crossing dataset are shown in Figure 6.

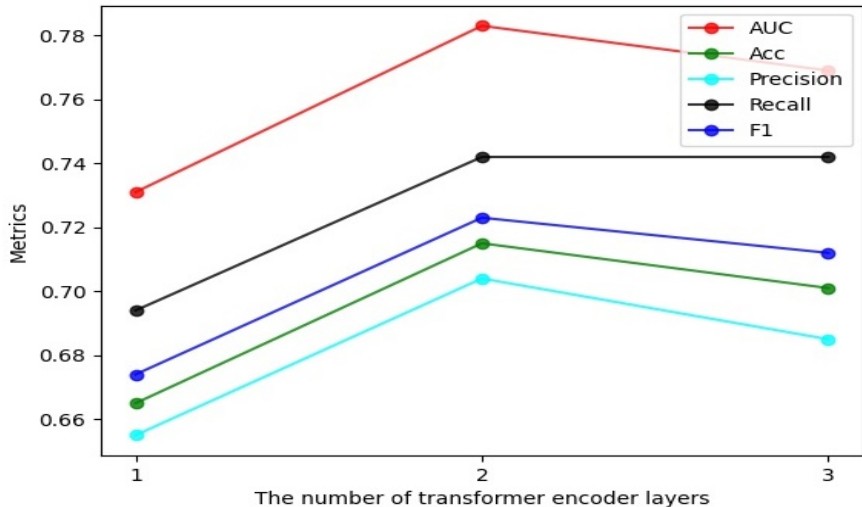

**Figure 6.** Effect of the number of transformer encoder layers on the experiment on Book-Crossing dataset.

In Figure 6, we clearly find that the number of transformer encoder layers has a significant impact on the final effect of the model. The metrics of the results using GCNTRec with a single-layer transformer encoder are at a very low level. When the number of layers of the transformer encoder is increased to 2, the metrics are significantly improved. However, the measurement indicators decreased to varying degrees when the number of layers was set to 3—and among them, the precision had a dramatic decline.

### 4.2.3. Effect of the Number of Heads in a Transformer Encoder Layer

Figure 7 shows the overall performance under different numbers of heads in a layer of transformer encoder on the Book-Crossing dataset. From the results, it is obvious that the more heads there are in a transformer encoder layer, the lower the values of AUC, Acc, Precision, and $F_1$ are. Only the recall obtained a greater value for 4 than 2. We think that it is normal to have slight fluctuations. Therefore, combining the changes of all the above metrics, we conclude that the smaller the number of heads is, the better the effect of our model is.

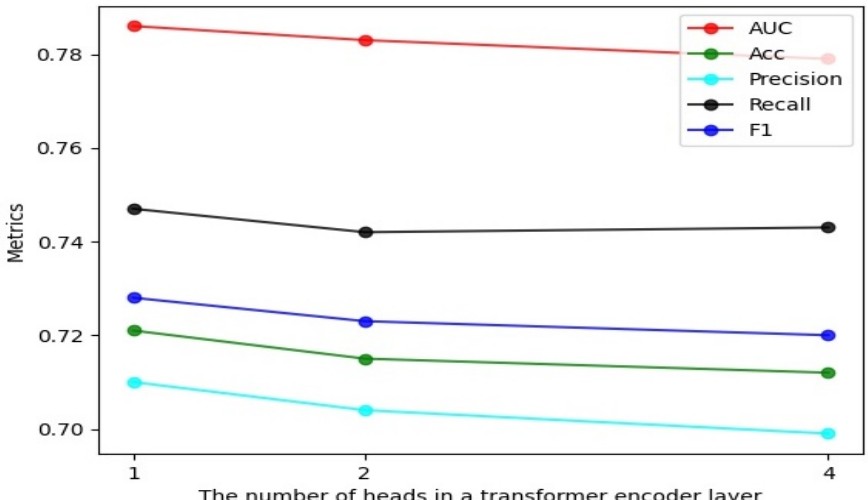

**Figure 7.** Effect of the number of heads in a transformer encoder layer on the experiment with the Book-Crossing dataset.

### 4.2.4. Effect of the Number of Layers of GCN

The model GCNTRec proposed in this paper finally uses a two-layer GCN structure. This parameter selection is based on the experimental comparative analysis of the number of layers of the graph neural network. On the dataset Steam-200k, the influence of the number of layers of the graph convolutional neural network on the experimental results is shown in Figure 8.

Obviously, when we use a layer of a graph convolutional neural network for feature extraction, the experiment produced poor results. Using a two-layer graph convolutional neural network is significantly better than a single-layer neural network. When the number of layers of the GCN is 3, the value of the metrics does not change significantly, but we can still see a slight decrease. Similarly, for a four-layer graph convolutional neural network, accuracy and recall are almost unchanged, but precision has a significant drop.

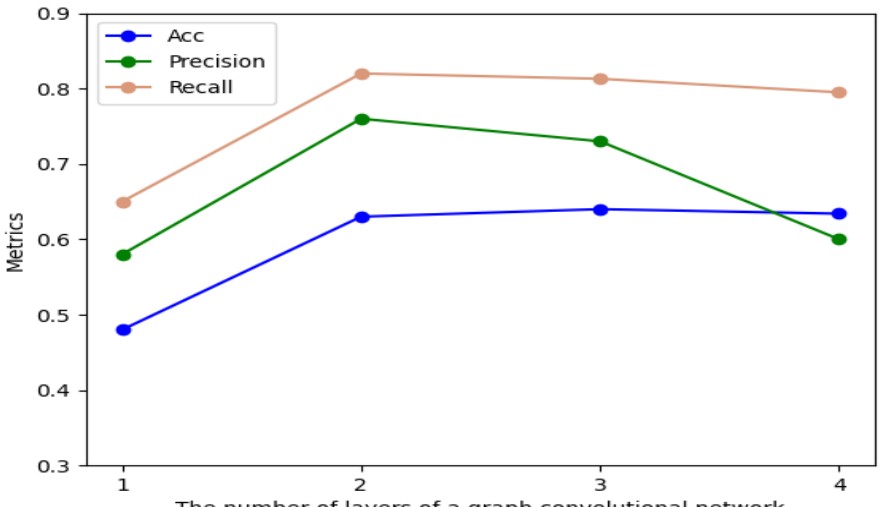

**Figure 8.** Effect of the number of layers of GCN on the experiment on the Steam-200k dataset.

## 5. Discussions

### 5.1. Why Does GCTNRec Work?

We proved the advantages of GCTNRec through experiments. Three advantages of GCTNRec which may explain its effectiveness in sequential recommendation are:

(1) **Modeling data from a heterogeneous perspective:** Most data scenarios in the real world are composed of multiple types of entities and various relationships between entities. Therefore, a heterogeneous graph used to describe each entity and the relationship between the entities contains rich information. Extracting the information from the heterogeneous graph will help us resolve the problem of sparse features of the entity itself.

(2) **Better sequence understanding through deep learning:** Unlike traditional techniques, GCNTRec learns sequences with GCN and the transformer encoder. Characteristics of items in sequences, such as adjacent items in graphs and item orders in sequences, are considered in these models. Therefore, it can better evaluate the relevance of sequences and items to predict the next item.

(3) **Degree coding further enriches the characteristics of items in sequences:** The degree of a node in a graph indicates the number of nodes adjacent to the node. The degree can reflect the influence level of a node in a graph. The higher the degree is, the more nodes are connected with the node. Therefore, the degree has a certain influence on the item feature representation for sequential recommendation tasks.

### 5.2. Threats to Validity

Our goal is to improve the feature extraction ability of sequences for sequential recommendation tasks. There is a threat of excessive data differences between different scenarios in the real world. To mitigate this threat, we try to select datasets from the real world with obvious differences in data characteristics for verification. We experimented with five datasets from five different scenarios in the real world. We believe that the threat of huge differences in data characteristics in different scenarios is not significant as we covered as many scenarios as possible. The most important goal of our experiments is to verify the feature extraction capability of GCNTRec when the target item itself has few obvious attributes.

## 6. Conclusions

In this paper, we propose a sequential recommendation method named GCNTRec. GCNTRec empirically investigates the advantage of fusing the transformer encoder model with degree encoding based on a graph neural network for sequential recommendation

tasks. We demonstrated that the transformer encoder model with degree representations outperforms state-of-the-art approaches by a large margin. In our future work, we plan to study the extraction of richer user features and apply the techniques in other tasks (e.g., supply chain forecasting tasks).

**Author Contributions:** Methodology, S.W., J.W.; software, X.L., X.K.; writing—original draft preparation, S.W., X.L., S.Z., J.G; writing—review and editing, S.W., X.L., X.K., J.Z.; project administration, J.W.; All authors have read and agreed to the published version of the manuscript.

**Funding:** This research received no external funding.

**Institutional Review Board Statement:** Not Applicable.

**Informed Consent Statement:** Not Applicable.

**Data Availability Statement:** Not Applicable.

**Conflicts of Interest:** The authors declare no conflict of interest.

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
