# Peer review of "Sequential Recommendation through Graph Neural Networks and Transformer Encoder with Degree Encoding"

_algorithms, doi:10.3390/a14090263_

Round 1

Reviewer 1 Report

This paper presents an approach based on graph neural networks and transformer-based encoders to learning effective item representation in a user historical behaviors sequence. The approach was evaluated with respect to some basslines using public datasets.

I have several comments and suggestions.

Introduction Section

* Clearly state the limitations of the existing works with respect to your problem

*Add a paragraph on the organization of the paper

Results Section

The results in Table 3 is not sufficient – only AUC is not good enough. Multiple metrics should be used. I recommend to use critical difference diagrams for each key metric (see https://github.com/hfawaz/cd-diagram for an implementation).

 It is necessary to justify the choices of hyperparameters (as much as possible).

Discussion Section

There is no discussion section. I recommend adding a discussion section, where the authors can discuss the implications of their findings and threats to validity.

Reviewer 2 Report

  1. The title numbering should start with 1, so Introduction is Section 1, Related work is Section 2, etc.
  2. Line 17 - It is stated: "Recommendation algorithms are generally combined with full items’ data ..." I recommend listing some examples of data that are commonly processed.
  3. Line 39 - I recommend specifying more extensively what the term "heterogeneous information networks" means.
  4. At the end of the Introduction, I recommend adding a paragraph where the paper organization is provided. 
  5. The abbreviation of  GCNTRec is defined multiple times. I recommend defining it once and then use only this abbreviation.
  6. Line 174 - I recommend specifying what kind of graphs you assume, i.e., directed/undirected, finite/infinite, etc? Please provide more details about the used graphs.
  7. Line 176 - This should be checked by the authors: "where K". Probably, "k" should be used instead of "K". 
  8. In the text, the term " a graph convolutional network" has been used several times. However, the authors defined its abbreviation "GCN". I recommend using it instead of the full name.
  9. Line 208 - I recommend changing the sentence in this line as follows: "We conduct our experiments on five public datasets, namely the BookCrossing dataset, Movielens-1m dataset, Steam-200k dataset, and Amazon Beauty dataset."
  10. Line 306 - It is stated: "From the results, it is obvious that the more heads in a transformer encoder layer, the lower the values of various metrics" However, the black line seems to take a greater value for 4 than for 2. This should be discussed in the paper.
  11. English should be revised as it contains several mistakes : 
    1. Line 6 - "s" is missing in which involve. Correct: which involves
    2. Line 5 - missing space in GCNTRec(Graph
    3. Line 39 - missing space in networks.It alleviates
    4. Line 35, 55 - "to learning" should be substituted with "to learn"
    5. Line 267 - Incorrectly used capital letter:  "...encoder, We..."
    6. also one of the authors' names (kou) does not start with a capital letter
    7. other mistakes such as missing articles, missing commas, missing spaces, bad word order

Round 2

Reviewer 1 Report

The authors have addressed my comments/concerns for the paper in this revision. 

Reviewer 2 Report

In my opinion, the paper can be accepted in its current form. However, there is some technical issue with section References. They are not included in the newest version.

This manuscript is a resubmission of an earlier submission. The following is a list of the peer review reports and author responses from that submission.